# Mitomycin C as an Anti-Persister Strategy against *Klebsiella pneumoniae:* Toxicity and Synergy Studies

**DOI:** 10.3390/antibiotics13090815

**Published:** 2024-08-28

**Authors:** Olga Pacios, Soraya Herrera-Espejo, Lucía Armán, Clara Ibarguren-Quiles, Lucía Blasco, Inés Bleriot, Laura Fernández-García, Concha Ortiz-Cartagena, María Paniagua, Antonio Barrio-Pujante, Belén Aracil, José Miguel Cisneros, María Eugenia Pachón-Ibáñez, María Tomás

**Affiliations:** 1Translational and Multidisciplinary Microbiology Research Group (MicroTM)-Microbiology Department, Biomedical Research Institute of A Coruña (INIBIC), A Coruña Hospital (CHUAC), University of A Coruña (UDC), 15006 A Coruña, Spain; olgapacios776@gmail.com (O.P.); luciaarman@gmail.com (L.A.); cibargurenq@gmail.com (C.I.-Q.); luciablasco@gmail.com (L.B.); bleriot.ines@gmail.com (I.B.); laugemis@gmail.com (L.F.-G.); cortizcartagena@hotmail.com (C.O.-C.); antonio99-bp@hotmail.com (A.B.-P.); 2Mechanisms of Antimicrobial Resistance Study Group (GEMARA) on Behalf of the Spanish Society of Infectious Diseases and Clinical Microbiology (SEIMC), 28003 Madrid, Spain; 3Institute of Biomedicine of Seville (IBiS), Virgen del Rocío University Hospital/CSIC/University of Seville, 41013 Seville, Spain; sherrera-ibis@us.es (S.H.-E.); maria.mapade@gmail.com (M.P.); jmcisnerosh@gmail.com (J.M.C.); mpachon-ibis@us.es (M.E.P.-I.); 4MEPRAM, Project of Personalized Medicine against Antimicrobial Resistance, 28029 Madrid, Spain; baracil@isciii.es; 5CIBER of Infectious Diseases (CIBERINFEC), Health Institute Carlos III, 28029 Madrid, Spain; 6Reference Laboratory of Antimicrobial Resistance, National Center of Microbiology, Health Institute Carlos III, Majadahonda, 28222 Madrid, Spain

**Keywords:** persistence, *K. pneumoniae*, repurposing, mitomycin C, lytic phage, cytotoxicity

## Abstract

The combination of several therapeutic strategies is often seen as a good way to decrease resistance rates, since bacteria can more easily overcome single-drug treatments than multi-drug ones. This strategy is especially attractive when several targets and subpopulations are affected, as it is the case of *Klebsiella pneumoniae* persister cells, a subpopulation of bacteria able to transiently survive antibiotic exposures. This work aims to evaluate the potential of a repurposed anticancer drug, mitomycin C, combined with the *K. pneumoniae* lytic phage vB_KpnM-VAC13 in vitro and its safety in an in vivo murine model against two clinical isolates of this pathogen, one of them exhibiting an imipenem-persister phenotype. At the same time, we verified the absence of toxicity of mitomycin C at the concentration using the human chondrocyte cell line T/C28a2. The viability of these human cells was checked using both cytotoxicity assays and flow cytometry.

## 1. Introduction

Infections caused by *Klebsiella pneumoniae* represent a rising concern in both hospital settings and the community, most of them being caused by multidrug-resistant (MDR) strains and difficult-to-treat isolates [1]. Many recent outbreaks have evidenced the need for effective treatments against antibiotic-resistant isolates, especially the carbapenemase-producing strains of *K. pneumoniae* (CRKp) [2,3,4,5].

Frequently underestimated, although clinically relevant, are the antibiotic-persister strains, a subpopulation of bacterial cells that enter into a dormant and latent metabolic state in which they are able to transiently survive antibiotic exposure while exhibiting sensitive minimal inhibitory concentration (MIC) values, which are below the breakpoints established for a specific antibiotic [6,7]. The persister subpopulation exits the metabolically dormant stage once the antibiotic pressure is removed, and is therefore responsible for recalcitrant infections and the chronification of these as well as usually being associated with biofilms [8,9]. In the clinical setting, persisters can survive in immunocompromised patients and in those in whom antibiotics did not effectively kill pathogenic bacteria, due to immune-evasion strategies. Examples of these are *Pseudomonas aeruginosa* and *Salmonella enterica* surviving in macrophages, together with uropathogenic strains of *Escherichia coli* [10,11,12,13]. Moreover, they can act as reservoirs for resistant mutants [14]. In vitro, persisters are characterized by a biphasic killing curve in the presence of the antibiotic, since the main susceptible population is rapidly killed, whereas the persister subpopulation has a slower killing time [6].

In this context, innovative strategies, such as the repurposing of drugs that have been approved by regulatory agencies with other therapeutic indications rather than antibacterial, together with the use of lytic bacteriophages (or phages), the natural predators of bacteria that specifically infect and kill their hosts, are of special interest [15]. The use of bacteriophages to treat bacterial infections in humans started a century ago, and many articles in the scientific literature report their successful therapeutic outcomes [16,17].

Concerning the repurposing strategy, mitomycin C (MMC), naturally produced by *Streptomyces caespitosus*, is an FDA-approved anticancer molecule used for the treatment of bladder, gastric, lung and pancreatic cancer, among others [18,19]. In bacteria, MMC passively enters the bacterial cell and gets reduced inside the cytoplasm, which enables its activation. MMC is an alkylating agent that covalently binds to DNA and induces interstrand crosslinking reactions between guanine adjacent bases in the minor groove of the double helix (5′-GC). These crosslinking reactions are resolved by the UvrABC system in cells in which this mechanism is undamaged and works properly. However, this is not the case for persister cells, where MMC has been proven to kill the subpopulation of these dormant cells in many different pathogens, such as in *Acinetobacter baumannii*, *E. coli* and *K. pneumoniae* [20,21]. As a chemotherapeutic agent, MMC is usually intravenously administered at concentrations ranging from 0.5 to 2 μg/mL, but concerns related to evaluating the toxicity of this molecule have arisen [22].

The repurposing of this compound as an antibiotic against *K. pneumoniae* infections, following the in vitro and in vivo approach set out by the *Galleria meillonella* model, was already performed by our research group [23]. In that work, subinhibitory concentrations of mitomycin C and the conventional carbapenem imipenem were combined with the lytic bacteriophage vB_KpnM-VAC13, and tested against (i) an imipenem-resistant *K. pneumoniae* isolate (MIC_IMP_ = 8 μg/mL), harboring the gene *bla_OXA-245_*; and (ii) an imipenem-persister isolate of this pathogen (MIC_IMP_ = 0.5 μg/mL). Based on previous reports that link lytic phage infection with entrance into the persister stage [24,25], this work aims to evaluate the safety of anti-persister MMC combined with the phage vB_KpnM-VAC13 in the in vivo murine model to broaden the applicability of this combinatorial approach. At the same time, we verified the absence of toxicity of MMC, at the concentration used against *K. pneumoniae*, using the human chondrocyte cell line T/C28a2, and assessed this with cytotoxicity assays and flow cytometry.

## 2. Results

### 2.1. Synergy Studies

#### 2.1.1. Optical Density Growth Curves and Viability Assay

Different outcomes in terms of growth were assessed for the three strains tested. In what concerns the imipenem-resistant isolate K2534 (ST437-OXA245), MMC at 3 μg/mL did not affect its growth for the first 8 h, exerting a slight activity from this time point onwards. The phage vB_KpnM-VAC13 completely inhibited the growth of K2534 for the first 8 hpi, and then the regrowth of a phage-resistant population occurred. In the case of this strain, no synergy was obtained for the combination between MMC and vB_KpnM-VAC13, although a slight delay in the arising of phage-resistant mutants is observed (Figure 1a).

Regarding the imipenem-persister strain K3325, an inverted effect in terms of susceptibility to MMC and to the vB_KpnM-VAC13 compared to K2534, was obtained: the MMC exhibited a much stronger bactericidal effect than in the OXA245-harboring isolate (Figure 1b). As already described in a previous work [23], the phage vB_KpnM-VAC13 did not exhibit a very strong lytic activity, and the culture regrew at 4 h post-infection. Importantly, the combination between MMC and vB_KpnM-VAC13 effectively killed the whole bacterial population, exhibiting in this case a strong synergism (Figure 1b).

Finally, for the reference strain ATCC^®^10031^TM^, the lytic phage exhibited a strong lytic effect for the first 10 h, but a regrowth was visible from this timepoint onwards. A complete arrest of the bacterial growth was obtained in the case of MMC alone, and in the combination of this compound with the lytic phage (Figure 1c).

In terms of viability, the enumeration of the colony-forming units (CFU) consistently reflected the results obtained with the spectrophotometer: non-statistically significant differences were obtained at 24 h for K2534 among the different conditions evaluated (Figure 1d), whereas for K3325, a synergistic effect was achieved in the presence of both MMC and the lytic phage, resulting in a 6-log decrease compared to the control counts. Interestingly, MMC alone only decreased the bacterial counts in 2-log, while the phage produced a 1-log reduction by itself, the effect of the combination being greater than the sum of the individual activities (Figure 1e). Finally, the ATCC^®^10031^TM^ strain revealed a drastic decrease in the CFU enumerated after exposure to MMC alone and combined with vB_KpnM-VAC13, whereas the phage alone did not produce a statistically significant reduction in the bacterial counts at 24 h compared to the control, which it is highly likely was due to the arising of phage resistance. 

#### 2.1.2. Frequency of Phage-Resistant Mutants

As previously evoked, the arising of phage-resistant mutants to the combined treatment herein evaluated (MMC and the lytic phage vB_KpnM-VAC13) might compromise the outcome of this approach. Therefore, we assessed the frequency of these mutants not only to the phage alone and combined with MMC, but also to the MMC by itself at the concentration used in the rest of the experiments, 3 mg/L.

We observed diverse patterns of resistance to the lytic phage exhibited by the different strains used, ranging from 9.9 × 10^−3^ ± 3.6 × 10^−3^ for K2534 to 7.3 × 10^−1^ for K3325 (already assessed in our previous work [23]), and 4.1 × 10^−5^ ± 1.5 × 10^−5^ in the case of the reference strain ATCC^®^10031^TM^ (Figure 2). In the case of the resistance shown in the presence of MMC at 3 mg/L, we obtained an expectedly high frequency of resistance in the case of K2534 (1.2 ± 0.2), which was 7.3 × 10^−5^ for K3325, and there were no viable colonies growing for the reference strain, hence the frequency of resistance to MMC of nearly zero. For K2534, the frequency of resistance to the combined MMC and the phage was not statistically significant compared to the phage resistance alone, whereas it was statistically different when comparing the frequency of MMC alone and in combination with the phage (*p*-value = 0.015). In the case of K3325, the frequency of resistance decreased to nearly zero in the presence of the combination, being statistically significant when compared to the phage resistance to vB_KpnM-VAC13 alone (*p*-value = 0.0035). Finally, the reference strain exhibited undifferentiated values of resistance to the MMC alone or combined with the phage (Figure 2).

#### 2.1.3. Assessment of the Cellular Respiration In Vitro

The cellular viability in the presence of MMC for every strain tested was comparable to that of the growth control, despite the differences in the growth patterns shown in the growth curves (Figure 3). Besides this, this assay has indicated that the lytic phage vB_KpnM-VAC13 at an MOI of 10 allowed normal metabolic activity for the two clinical strains (a and b), differently from the effect observed in the reference strain (c), in which it significantly reduced the metabolic activity (*p* < 0.001). For this strain, but interestingly also for the persister isolate K3325, combinations of phage and MMC were able to significantly reduce the metabolic activity of bacteria compared to the agents separately.

### 2.2. Toxicity Studies

#### 2.2.1. In Vitro: Metabolic Activity and Apoptosis Study

Next, we performed a cellular cytotoxic assay using the MTT agent (3-[4,5-dimethylthiazol-2-yl]-2,5-diphenyl tetrazolium bromide), based on the colorimetric reduction of the tetrazolium salt to purple formazan crystals. This reaction allows the measurement of metabolically active cells, as an indicator of cellular proliferation and metabolism. We visually assessed the conversion of MTT to purple formazan after the addition of DMSO (Figure 4a), and calculated the percentage of viability of T/C28a2 cells under the concentrations evaluated (Figure 4b). Low percentages of viability (around 35%) were obtained for 20 and 10 µg/mL MMC, showing non-statistically significant differences between these two concentrations, whereas an MMC at 8 µg/mL led to a viability of 43% on average, which represented a statistically significant decrease compared to the viability at MMC = 6 µg/mL (around 55%). Interestingly, at 3 µg/mL (the concentration used for the other experiments here reported), a statistically significant increase was observed compared to the viability at a concentration of 6 µg/mL, reaching 75% (Figure 4b).

These results were confirmed by flow cytometry, which revealed a statistically significant increase in the percentage of live cells (PI and annexin-FITC negative) treated with 3 µg/mL of MMC compared to 10 µg/mL, and a significant decrease in the percentage of apoptotic cells when comparing these two concentrations (*p* < 0.001, Figure 4c). Importantly, the percentage of apoptotic cells was not statistically different comparing absence of MMC and MMC at 3 µg/mL. Finally, no difference in the percentage of necrotic cells was observed among the different concentrations of MMC tested.

Representative graphs of each condition (one technical replicate) obtained with the software CytExpert 2.0 are shown in Figure 4d.

#### 2.2.2. In Vivo Toxicity Assay: Murine Model

Acute toxicity was assessed in a murine model after the intraperitoneal injection of a single dose of 2.4, 1.7 and 0.52 mg/kg MMC concentrations, and the lytic bacteriophage vB_KpnM-VAC13 at MOIs of 0.50, 1 and 10. No systemic signs of pain or weight loss were observed.

Cumulative toxicity was assessed using the highest concentrations of MMC and the bacteriophage—that is, MMC at 2.4 mg/kg/ip/24h and vB_KpnM-VAC13 at 10 MOI/ip/24h—for 72 h. After evaluation at 7 days post-treatment, no systemic signs of pain or weight loss were observed.

## 3. Discussion

The repurposing or repositioning of drugs is useful when re-evaluating the therapeutic indications of a pharmaceutic product [26,27]. At a time when new antibiotics are no longer effective, and resistance and persistence to them are increasing, the use of drugs that have already been developed, tested and approved for use in humans with therapeutic indications other than antibacterial could be increased as a promising alternative [28]. Furthermore, it is important to note that antibiotics induce a disturbance in the healthy microbiota, with a concomitant reduction in bacterial species diversity, altered metabolic activity, and the selection of antibiotic-resistant microorganisms [29].

Several studies have already discussed MMC as a readily applicable treatment for clinical infections, regarding its bactericidal effects against diverse pathogens [19,20,23,30,31]. In this work, we have further analyzed the possible synergism between the lytic bacteriophage vB_KpnM-VAC13 in combination with MMC in two different clinical strains of *K. pneumoniae* and in one reference strain. Furthermore, we verified the absence of toxicity in the repurposed anticancer drug mitomycin C both in vitro and in healthy immunocompetent mice, together with an in vivo toxicity assay with the lytic phage vB_KpnM-VAC13 at its highest concentration.

A synergistic effect between MMC and the lytic phage on the *K. pneumoniae* imipenem-persister isolate K3325 (Figure 1 and Figure 2) was observed, verified by a decrease of more than 2 log of CFU/mL in the presence of the combination compared with the most active single agent (Figure 1e) [32,33]. The resistant isolate K2534 did not show any detriment to its growth in the presence of MMC, which is consistent with its high MIC to this compound (25 mg/L), as already determined in a preliminary study published by our research group [23]. On the other hand, the persister isolate K3325 showed the highest inhibition of its growth in the presence of MMC, consistent with its MIC value of 6.25 mg/L [23]. Besides this, the reference strain ATCC^®^10031^TM^ showed a heavily detrimental effect on its growth under all the conditions evaluated: phage alone, mitomycin C alone and the combination of these two (Figure 1c,f).

The different growth curves shown by K2534 and K3325 in the presence of this anticancer molecule, alone and combined, could be explained by the distinctive working mechanism of the nucleotide excision repair (NER) UvrABC system, as well as other defense systems that resolve double-strand breaks provoked by MMC [34]. As K2534 is not a persister isolate, it is highly likely that the UvrABC system works properly and resolves the DNA crosslinking reactions that MMC induces in its genome, as previously observed [20,21,35], which would explain the absence of bactericidal activity under the conditions in which MMC was added for this isolate.

All in all, we have concluded that MMC at 3 μg/mL exhibited a bactericidal activity both in vitro and in vivo (even a synergistic effect when combined with a lytic phage), at least in the case of bacterial persister cells in which NER mechanisms are altered.

Concerning the infection with the phage vB_KpnM-VAC13, different patterns of growth inhibition were observed in vitro (Figure 1). In the case of K2534, OD growth curves revealed a complete inhibition of its growth for the first 8 hpi, and then a regrowth was visible, probably due to the rapid development of phage resistance mutations [36]. This would explain the high frequency of resistant mutants that was assessed for this strain (Figure 2). Interestingly, vB_KpnM-VAC13 did not produce the same effect on the K3325 isolate, being unable to produce visible lysis when assessed with the spectrophotometer (Figure 1b), and this was confirmed with the unusually high frequency of phage-resistant mutants exhibited by this strain (Figure 2). However, the combination of MMC and the phage successfully reduced the frequency of resistance for both K3325 and the reference strain; differently but expectedly, the K2534 isolate did not show any statistically significant difference in the resistance rates among the three conditions evaluated.

Even if there are toxicity concerns associated with the administration of MMC, the concentrations at which this anticancer agent exerts bactericidal effects are similar to the therapeutic ones that have been established for cancer treatments: MMC has been safely administered at 2 μg/mL intravenously, and up to 400 μg/mL topically [18,21]. Regarding the toxicity issues that this molecule entails, here, we have performed both acute and cumulative toxicity assays, using healthy immunocompetent mice, together with an in vitro assay using cultured human chondrocytes belonging to the cell line T/C28a2.

We then used the human chondrocytes cell line T/C28a2 and exposed it to several concentrations of MMC for 24 h, measured the metabolic activity of the cells and quantified the populations that were alive, dead, or had entered an apoptotic state by flux cytometry (Figure 4). Importantly, at this concentration, a high percentage of viability was assessed for human chondrocytes, together with a low level of apoptosis induction, verifying the absence of toxicity of this compound. As concerns the in vivo model, we did not observe any systemic signs of pain or weight loss for the concentrations evaluated, neither for the acute toxicity experiment nor for the cumulative toxicity at 72 h. This confirms the safety of MMC and vB_KpnM-VAC13 at these concentrations in healthy female immunocompetent mice, even if further experiments will be needed to confirm these results. Other works interrogating the potential toxicity of mitomycin C using animal models (either rats or mice) have been published in this regard, and similar outcomes as the ones herein described were reported: in 2005, Babu et al. evaluated the ototoxicity of mitomycin C and observed that at concentrations < 0.20 mg/mL, no alterations in the auditory brainstem threshold were noted [37]. Furthermore, the use of adjuvants, i.e., high-molecular-weight polymers such as icodextrin [38] or pegylated lysosomes [39], has been proven to drastically reduce the toxicity of mitomycin C in rodents. Finally, these polymers could contribute to enhancing the stability of the complex mitomycin C-lytic phage, even if no chemical interaction or neutralization is expected between these two and no works have reported any concerns in this regard. However, further experiments that empirically address this question in a specific clinical environment might be required.

Taking into consideration that the discovery and development of new antibiotics are currently limited and, in most cases, unsuccessful, the repurposing approach is gaining interest. As it represents a notable reduction in the time, risks and costs associated with the production of new drugs, since the repositioned molecules have already been approved by regulatory agencies and proven to be safe and efficacious, efforts must be made in order to find synergistic relations between repurposed compounds and other antibacterials, such as lytic phages [40,41]. This work represents one example of an exciting synergism between an anticancer compound and a lytic phage, proven to be effective against persister subpopulations of *K. pneumoniae* and, importantly, exhibiting no toxicity against human cells and healthy immunocompetent mice.

## 4. Materials and Methods

### 4.1. Bacterial Strains, Phage and Growth Culture Conditions

*K. pneumoniae* clinical strains K2534 (imipenem-resistant) and K3325 (imipenem-persister) were isolated and stored at the National Center of Microbiology (CNM, Madrid, Spain). *K. pneumoniae* K2534 (ST437-OXA245) was isolated from a rectal sample, whereas K3325 was isolated from a blood sample. *K. pneumoniae* subsp. *pneumoniae* reference strain ATCC^®^10031^TM^ was also employed for several experiments. All the strains were cultured in Luria-Bertani (LB) growth medium (1% tryptone, 0.5% NaCl and 0.5% yeast extract) at 37 °C and, most of the time, in shaking conditions (180 rpm).

The lytic bacteriophage vB_KpnM-VAC13 was isolated from sewage water and phenotypically and genomically characterized by our group in recent works [23,42]. In general, for the propagation of lytic bacteriophages, we used saline magnesium (SM) buffer. The composition of the SM buffer is as follows: NaCl 100 mM, MgSO_4_ 7H_2_O 8 mM, Tris–HCl (pH7.5) 50 mM, made up to 1000 mL with distilled water. However, for subsequent dilutions prepared for in vivo experiments, saline buffer 0.9% was used. MMC was purchased at SigmaAldrich^®^ (Madrid, Spain) and diluted in deionized water or filter-sterilized saline buffer.

The characteristics of the *K. pneumoniae* strains and the lytic phage used throughout this work are reported in Table 1.

### 4.2. Optical Density Growth Curves

Overnight cultures of *K. pneumoniae* clinical strains K2534 and K3325 and the reference strain *K. pneumoniae* ATCC^®^10031^TM^ were cultured in LB medium in 96-well flat-bottom plates (200 μL final volume) in the presence of the lytic bacteriophage vB_KpnM-VAC13 at multiplicities of infection (MOI) of 10, in the presence of 3 μg/mL of MMC, or with both together. In every case, MMC was added 1 h post-infection (hpi) to allow entrance into the persistent state. A row of LB exclusively inoculated with bacteria was used as the growth control, while the other two rows containing either non-inoculated LB or MMC at 3 μg/mL were included as blanks. The plates were incubated with continuous shaking and read using an EPOCH Microplate Reader (BioTek^®^, Winooski, VT, USA) at an optical density of 600 nm (OD600 nm), measured every 15 min for 24 hpi.

### 4.3. Time-Kill Curves Assay

The same setup as described in the aforementioned section was used for the viability assay. At 24 h, the cultures were serially diluted in saline buffer and plated in duplicate on LB-agar plates (1% tryptone, 0.5% NaCl and yeast extract, 2% agar). Briefly, the curves started with an initial inoculum of 10^7^ CFU/mL, and samples were taken at 0 and 24 h. Bactericidal activity was defined as a decrease of ≥3 log10 CFU/mL from the initial inoculum, whereas synergy was defined as a decrease of ≥2 log10 CFU/mL for the combination compared with the most active single agent [32].

### 4.4. Frequency of Resistant Mutants

Overnight cultures of three different clones belonging to *K. pneumoniae* isolates K2534, K3325 and ATCC^®^10031^TM^ were diluted 1:100 in LB broth medium until an optical density equivalent to 10^7^ CFU/mL was reached. Dilutions of the cultures were performed in saline buffer, then 100 µL were plated on LB-agar plates containing 3 mg/L of MMC to assess the resistance arising to this molecule alone. To calculate the frequency of phage-resistant mutants to the phage vB_KpnM-VAC13 alone or combined with MMC, 100 µL of a solution containing 10^9^ plaque-forming units (PFU)/mL was added to 100 µL of the diluted cultures, then plated using the top-agar method (or double-layer method) onto TA-agar plates (1% tryptone, 0.5% NaCl, 1.5% agar), or onto MMC-containing TA plates (supplemented with MMC at 3 mg/L). The top-agar method has been previously described in [43,44], and the protocol of determination of the phage resistance frequency is based on previous works as well [45].

### 4.5. Respiration Assay Using the Tetrazolium Salt WST-1

*K. pneumoniae* clinical strains K2534 and K3325 and the reference strain *K. pneumoniae* ATCC^®^10031^TM^ were cultured in LB medium in flat-bottom 96-well plates (200 μL final volume) in the presence of either the phage at MOI = 10, MMC (3 μg/mL), or both (MMC added 1 hpi). The reagent WST-1 (Roche^®^, Pleasanton, CA, USA), a tetrazolium salt that produces a color change in the medium when reduced by the presence of NAD(P)H, was diluted 1:100 in every well, including the growth control and the blanks. NADPH is considered an indicator of bacterial metabolic activity. The plates were incubated without shaking and read within an EPOCH Microplate Reader (BioTek^®^). The OD at a wavelength of 480 nm was measured every 15 min during 24 hpi for K2534 and ATCC^®^10031^TM^ and at 12 hpi for K3325.

### 4.6. Cytotoxicity Assay Using Human Chondrocytes T/C28a2 Cell Line

#### 4.6.1. MTT-Cytotoxic Assay

Cell viability was determined using an MTT-based cytotoxicity assay on human cells, specifically the T/C28a2 chondrocyte cell line. MTT (3-[4,5-dimethylthiazol-2-yl]-2,5-diphenyl tetrazolium bromide) produces purple formazan from the mitochondrial enzymes of viable cells. Cultured T/C28a2 chondrocytes at a concentration of 6500 cells/well were seeded in a 96-well cell culture plate (Corning^®^, Corning, NY, USA) and allowed to form a monolayer for 24 h. After visually determining that the cells had reached a confluence of around 80% (medium cell density), they were exposed to 200 μL of DMEM High Glucose-Pyruvate medium (Gibco^®^, Waltham, MA, USA) supplemented with 10% fetal bovine serum (FBS, Gibco^®^) and 1% penicillin/streptomycin, containing various concentrations of MMC (firstly, 0.3 to 300 μg/mL were assessed in 10-fold, then the range was restricted to 3, 6, 8, 10 and 20 mg/L of MMC) for 24 h and 48 h. A non-supplemented medium (DMEM + 0% FBS) was included as a control to verify that the observed effect was indeed due to the presence of the drug.

Then, 200 μL of MTT diluted in DMEM was added to the wells. After 3 h incubation at 37 °C in a 5% CO_2_ incubator, the dye solution was carefully removed, and the crystals of purple formazan were solubilized with 100 μL pure DMSO (Sigma Aldrich^®^, Madrid, Spain). The plate was then incubated for 15 min at room temperature on a shaker, and absorbance at 570 nm and 630 nm (reference) was measured using a NanoQuant microplate reader. The experiment was performed using six technical replicates from three biological replicates.

#### 4.6.2. Apoptosis Study Using Flow Cytometry

Cultured human T/C28a2 chondrocytes, kindly provided by Dr. Mercedes Fernández Moreno and Dr. Carlos Vaamonde García (INIBIC) were seeded in a 48-well culture plate (Corning^®^, USA) at a concentration of 15,000 cells/well and allowed to form a monolayer for 24 h. After checking that the cells had reached a medium cell confluence, they were exposed to 300 μL of DMEM medium supplemented with 10% FBS and 1% penicillin/streptomycin, containing a range of MMC concentrations (0—only supplemented with DMEM, 3, 6 and 10 mg/L), and were incubated for 24 h at 5% CO_2_ and 37 °C. Six wells were included as technical replicates, and the experiment was performed using biological replicates.

After 24 h of incubation with the MMC, the medium was aspirated and kept in previously labeled Eppendorf tubes to take into consideration any naturally occurring dead cells. The wells were rinsed with filtered saline buffer, and 100 µL of 2× Trypsin-EDTA 0.5% (Gibco^®^, USA) was added to each well. The plate was incubated at 37 °C for 5 min to induce the detachment of the chondrocytes. Meanwhile, the Eppendorf tubes were centrifuged at 1500 rpm for 5 min and the supernatants were discarded. The trypsinized cells were added to the corresponding pellets obtained after the last centrifugation, and the tubes were again centrifuged for 5 min at 1500 rpm. After removing the supernatants once again, the cells were washed with 300 µL of saline buffer, and this step was repeated once more. Finally, pelleted cells were resuspended with the Binding Buffer (1×) of the “Dead Cell Apoptosis Kit with Annexin V FITC & Propidium Iodide for Flow Cytometry” (Invitrogen^TM^, Carlsbad, CA, USA) containing 1 µL of propidium iodide (PI) at 1.5 mM in deionized water, and 5 µL of annexin V-FITC (25 mM HEPES, 140 mM NaCl, 1 mM EDTA pH 7.4, 0.1% BSA) per reaction. Half of the tubes were resuspended in saline buffer without any PI or annexin V-FITC, and they were used as controls to define the gates at the flow cytometer CytoFLEX S (Beckman Coulter, Brea, CA, USA).

### 4.7. Acute and Cumulative of MMC and Lytic Bacteriophage vB_KpnM-VAC13 in Healthy Female C57BL6/J Mice

Immunocompetent C57BL/6J mice weighing 20 g were used. The mice had murine pathogen-free sanitary status and were assessed for genetic authenticity. This study was conducted in accordance with the recommendations of the Guide for the Care and Use of Laboratory Animals (National Research Council, Guide for the Care and Use of Laboratory Animals, The National Academies Press 2011). The experiments were approved by the Committee on the Ethics of Animal Experiments of the Regional Ministry of Agriculture, Livestock, Fisheries and Development, Spain (04/07/2022/099).

For acute toxicity, groups of 6 healthy female mice that were 7 weeks old were intraperitoneally (ip) treated with a single dose of MMC at concentrations of 2.4, 1.7 and 0.52 mg/L and vB_KpnM-VAC13 at MOI of 0.5, 1 and 10. After the doses and over seven days, the following indicative signs of pain were assessed: reduced water (dehydration) or food intake; isolation; self-mutilation; tremors/spasms; dyspnea; physical activity (increased/reduced); chromodacryorrhoea; muscle stiffness; piloerection; teeth grinding; or weight loss. For the cumulative toxicity, groups of 6 healthy C57BL/6J female mice received for 72 h the highest dose associated with no signs of toxicity in the acute toxicity studies previously evaluated, which was 2.4 mg/kg/ip/q24h and MOI10/ip/q24h for MMC and/or vB_KpnM-VAC13, respectively. The same systematic signs of pain, listed above, were monitored.

## Figures and Tables

**Figure 1 antibiotics-13-00815-f001:**
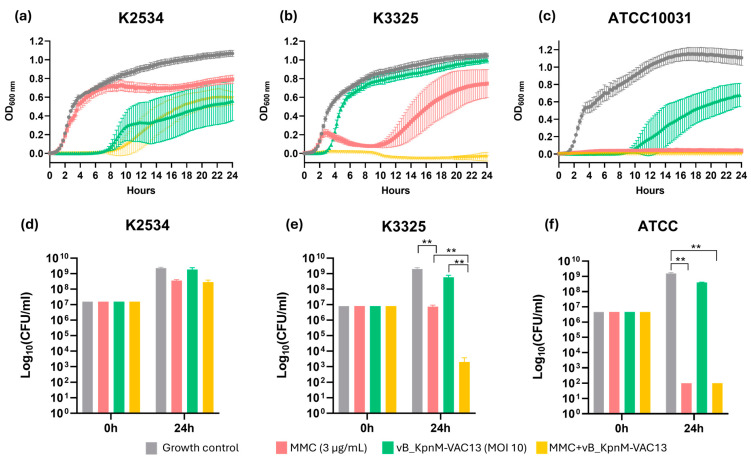
(**a**–**c**) Optical density growth curves performed with a microtiter plate reader and incubator at a wavelength of 600 nm every 15 min, using *K. pneumoniae* clinical strains K2534 (**a**), K3325 (**b**) and the reference strain ATCC^®^10031^TM^ (**c**), all infected with vB_KpnM-VAC13. (**d**–**f**) Viability of these three strains at the tested conditions, assessed after CFU enumeration. Only statistically significant comparisons have been depicted (** represents a *p*-value < 0.01).

**Figure 2 antibiotics-13-00815-f002:**
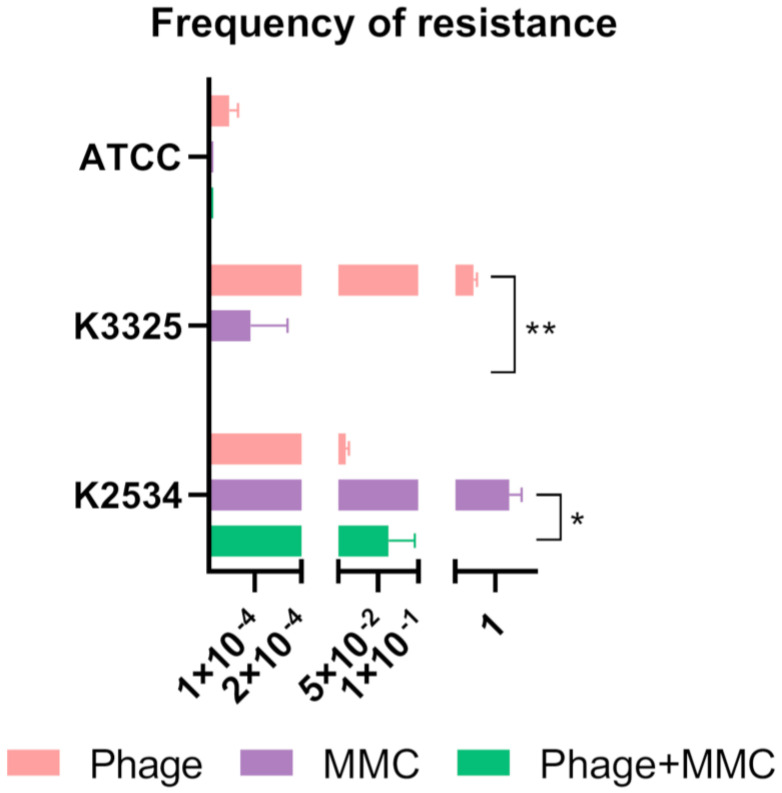
Frequency of resistant mutants in the presence of the phage alone, mitomycin C (MMC) at 3 mg/L and both combined; * represents *p*-values between 0.05 and 0.01; ** represents *p*-values between 0.01 and 0.001.

**Figure 3 antibiotics-13-00815-f003:**
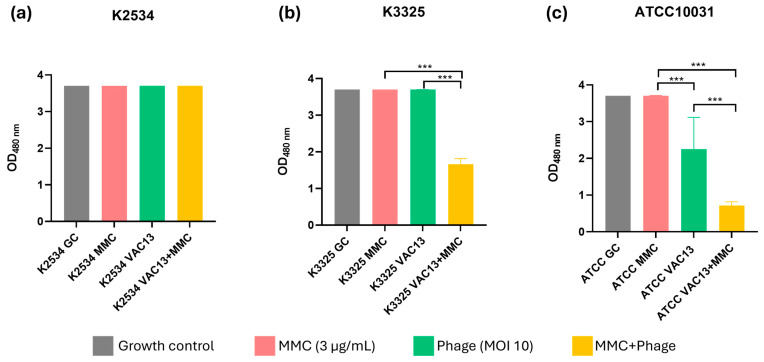
Respiration assay of *K. pneumoniae* clinical strains K2534 (**a**), K3325 (**b**) and the reference strain ATCC^®^10031^TM^ (**c**), using the tetrazolium salt WST-1 (an indicator of NADP^+^ reduction). *** represents *p*-values lower than 0.001.

**Figure 4 antibiotics-13-00815-f004:**
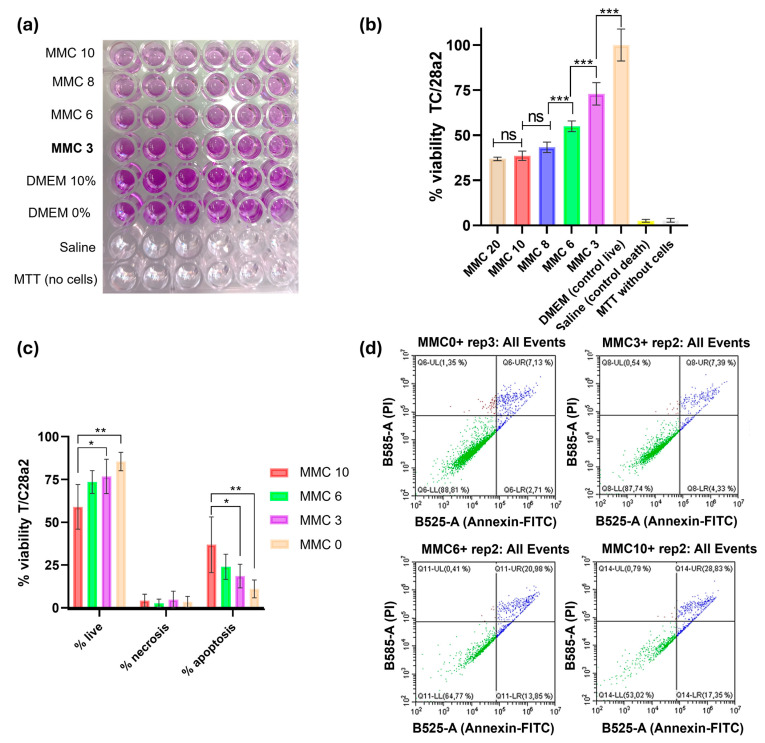
In vitro evaluation of the toxicity of MMC using the human chondrocyte cell line T/C28a2. (**a**) A 96-well microplate containing T/C28a2 cells after MTT-incubation and DMSO addition. (**b**) MTT cytotoxic assay represented as the percentage of viability after measurement of the OD_570nm_ at 24 h of incubation with the corresponding concentrations of MMC. *** represents *p*-values lower than 0.001. (**c**) Viability assay using the Live-Dead Apoptosis Kit (Thermo Scientific, Rockford, IL, USA), with propidium iodide and annexin V-FITC and using the flow cytometer CytoFLEX (Beckman Coulter, Brea, CA, USA). * represents *p*-values between 0.05 and 0.01; ** represents *p*-values between 0.01 and 0.001. (**d**) Representative graphs after flow cytometry analysis using the CytExpert 2.0 software (PI—propidium iodide; FITC—fluorescein isothiocyanate).

**Table 1 antibiotics-13-00815-t001:** List of strains and bacteriophages used in this study.

*K. pneumoniae* Clinical Strain	Sequence Type	Carbapenemase	Biological Origin	GenBank Accession Number	Reference
K2534	ST437	OXA-245	Rectal	WRXG00000000.1	[23]
K3325	ST42	-	Blood	SAMEA3649525	[23]
ATCC	ST131	-	Commercial		-
***K. pneumoniae* lytic Bacteriophage**	**Morphotype**	**Genus**	**Genome Size (bp)**	**GenBank Accession Number**	**Characterization (Reference) **
vB_KpnM-VAC13	*Myoviridae*	*Slopekvirus*	174,826	MZ322895.1	[23]

## Data Availability

Genomic data on the clinical strains and the lytic phages used in this work can be found in GenBank (NCBI) under these accession numbers: WRXG00000000.1, SAMEA3649525 and MZ322895.1, respectively.

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
