# Peer review of "Mitomycin C as an Anti-Persister Strategy against Klebsiella pneumoniae: Toxicity and Synergy Studies"

_antibiotics, 2024, doi:10.3390/antibiotics13090815_

Round 1

Reviewer 1 Report

Comments and Suggestions for Authors

The manuscript presents an interesting approach regarding the problems of antibiotic resistance by reusing for this purpose certain medicines established for other diseases (in the given situation of an antitumor medicine -Mitomycin C).  In my opinion the manuscript presents a great interest for the readers of this journal and I recommend the publication after a minor revision.

As comments/sugestions:

1. What can be said about the stability in biological environments of the Mitomycin C-K. pneumoniae lytic phage vB_KpnM-VAC13 mixture?

2. What solvent was used to test the in vitro/in vivo activity? What is its concentration in the final solution (DMSO itself has antibacterial activity, therefore in the tested solutions containing it it must not be in a concentration higher than 1%)

3. I suggest a broader description of the in vivo toxicity assay - information related to liver, kidney, blood parameters to better understand the degree of toxicity of this mixture

Pictograma Verificată de Comunitate      

Author Response

The first, thanks for this nice review of our work. Following the answers to the questions:

1. What can be said about the stability in biological environments of the Mitomycin C-K. pneumoniae lytic phage vB_KpnM-VAC13 mixture?

Stability of the combination is thought to be maintained when both agents are mixed, as no chemical interaction or neutralization is expected between a lytic phage and the molecule of mitomycin C and no works have reported this issue. Information about this has been added in lines 296-300.

2. What solvent was used to test the in vitro/in vivo activity? What is its concentration in the final solution (DMSO itself has antibacterial activity, therefore in the tested solutions containing it it must not be in a concentration higher than 1%)

-For in vitro propagation of bacteriophage saline magnesium (SM) buffer was used. The composition of SM buffer is: NaCl 100 mM, MgSO4. 7H2O 8 mM, Tris–HCl (pH7.5) 50 mM, made up to 1000 mL distilled water. However, for subsequent dilution prepared for in vivo experiments saline buffer 0.9% was used. (added in line 322-326).

-Moreover, MMC was diluted in deionized water or filter-sterilized saline buffer for the in vitro activity. (Lines 326-327)

-Finally, DMSO was never used for antibacterial experiments, but only for the in vitro cytotoxicity assay using the T/C28a2 human chondrocytes. Pure DMSO (100%) was used to solubilize the purple formazan crystals, formed after cellular respiration exclusively on metabolically active cells (added in line 397).

3. I suggest a broader description of the in vivo toxicity assay - information related to liver, kidney, blood parameters to better understand the degree of toxicity of this mixture

More data related to the in vivo toxicity assays, obtained from other works, has been summarized in lines 290-300.

Reviewer 2 Report

Comments and Suggestions for Authors

Drug repurposing involves utilizing an existing therapeutic agent for new indications beyond its original purpose. This approach is attractive due to potentially lower development costs and shorter timelines compared to developing new drugs. In this study, Tomás and colleagues assess the potential of repurposing the anticancer drug mitomycin C in combination with the K. pneumoniae lytic phage vB_KpnM-VAC13. They evaluate its effectiveness in vitro and its safety in an in vivo murine model against two clinical isolates of this pathogen. Additionally, the authors determined the acute toxicity of mitomycin C in a murine model after intraperitoneal injection. The viability of these human cells was assessed using cytotoxicity assays and flow cytometry. Overall, the manuscript is well-written and well-organized, and the references are adequately presented. Therefore, I support its publication in Antibiotics.

Additional comment:

It is suggested to include the in vivo data (combination therapy) to further improve the quality of this work.

Author Response

Thanks for the comments that improved our work. Following the answer of it.

It is suggested to include the in vivo data (combination therapy) to further improve the quality of this work.

More data related to the in vivo toxicity assays, obtained from other works, has been summarized in lines 290-300.